# Non-Deterministic Behavior of Thompson Sampling with Linear Payoffs and How to Avoid It

**Doruk Kilitçioğlu**                                               *doruk.kilitcioglu@fmr.com*
*Fidelity Investments*
*245 Summer St*
*Boston MA 02210*

**Serdar Kadıoğlu**                                               *serdar.kadioglu@fmr.com*
*Fidelity Investments*
*245 Summer St*
*Boston MA 02210*

**Reviewed on OpenReview:** *https://openreview.net/forum?id=sX9d3gfwtE*

## Abstract

Thompson Sampling with Linear Payoffs (LinTS) is popular contextual bandit algorithm for solving sequential decision making problems. While LinTS has been studied extensively in the academic literature, surprisingly, its behavior in terms of reproducibility did not receive the same attention. In this paper, we show that a standard and seemingly correct LinTS implementation leads to non-deterministic behavior. This might go unnoticed easily, yet impact results adversely. This calls the reproducibility of papers that use LinTS into question. Further, it forbids using this particular implementation in any industrial application where reproducibility is critical not only for debugging purposes but also for the trustworthiness of machine learning models. We first study the root cause of the non-deterministic behavior. We then conduct experiments on recommendation system benchmarks to demonstrate the impact of non-deterministic behavior in terms of reproducibility and downstream metrics. Finally, as a remedy, we show how to avoid the issue to ensure reproducible results and share general advice for practitioners.

## 1 Introduction

Contextual multi-armed bandit (MAB) algorithms are powerful solutions for online decision making problems in many areas of Information Retrieval (IR) including online advertisement (Schwartz et al., 2017) and personalized recommendation systems (Li et al., 2010). In this setting, an agent makes sequential decisions under uncertainty. In contextual bandits, the agent also observes a feature vector (*context*) associated with each decision. Given the contextual information, the agent selects an *arm* and receives a *reward*. The mechanism that determines the reward is unknown to the agent, and the reward of the chosen arm can only be observed at runtime. The agent should carefully balance exploration and exploitation to maximize cumulative reward (Auer et al., 2002).

The main methods that target the exploration-exploitation trade-off fall into two families with several variants: upper confidence bound (UCB) (Li et al., 2010) and Thompson Sampling (TS) (Thompson, 1933). While the former selects the arm with the highest UCB, the latter is a Bayesian approach that maintains a posterior distribution.

Among the bandit policies, Thompson Sampling with linear payoffs (LinTS), has attracted significant attention in the bandit literature. The original LinTS paper (Agrawal & Goyal, 2013) from 2013 has more than 600 citations[1] including many from the IR and recommender systems literature (Abeille & Lazaric, 2017;

---

[1]https://scholar.google.com/scholar?cites=16506820398491305928

Garcelon et al., 2020; Dimakopoulou et al., 2019; Gutowski et al., 2019; 2021). Moreover, LinTS is implemented in several machine learning libraries including the STRIATUM Python library from National Taiwan University (Lin et al., 2016), STREAMINGBANDIT REST service from Nth Iteration Labs (Kruijswijk et al., 2016), the CONTEXTUAL R package from Tilburg University, DEEP BAYESIAN BANDITS Tensorflow-based Python library from Google Brain with a variation that uses Bayesian Linear Regression (Riquelme et al., 2018), and finally, as part of MABWISER (Strong et al., 2021; 2019), our own Python library for contextual multi-armed bandits from Fidelity Investments[2].

The LinTS algorithm is popular for its Bayesian approach to exploration and empirical gains in performance over other contextual bandit algorithms (Li et al., 2010). The original LinTS paper from Microsoft Research by Agrawal & Goyal (2013) provides an elegant proof for the non-trivial regret bound of the algorithm using martingale-based analysis. Unfortunately, the paper does not provide an implementation or any computational results to demonstrate algorithm performance. As such, the reproducibility of this work remained unknown to date. In this paper, we put LinTS to a test.

Python is one of the most popular programming languages in machine learning research and data science practice. When implementing the LinTS algorithm in Python as part of our personalization efforts at Fidelity, we discovered that a standard and seemingly correct implementation leads to subtle non-determinism. Such behavior might easily go unnoticed, yet it affects the predictions adversely and alters conclusions drawn. Non-deterministic behavior has several undesirable consequences. First, it casts a shadow on the reproducibility of papers and applications where this implementation is used. Second, it cannot be part of any industrial application where reproducibility is critical not only for reproducing recommendation results and debugging, but also for potential external audits required by law and trustworthiness of algorithms.

In this paper, we first show that the main issue behind the non-deterministic behavior stems from the multivariate normal distribution functionality in NUMPY (Harris et al., 2020). The NUMPY package is fundamental in numeric computation, hence, it is reasonable to assume that LinTS implementations will depend on NUMPY. In fact, this non-deterministic multivariate normal distribution functionality is used in existing LinTS implementations found in STRIATUM[3], STREAMINGBANDIT[4], and DEEP BAYESIAN BANDITS[5], as can be seen in their Github repositories. We find the underlying cause to be prevalent across the open source community[6]. This casts a shadow on the reproducibility of *any work* relying on these libraries, and on *any research* that relies on NUMPY's multivariate normal sampling.

Unfortunately, such non-deterministic behavior is not documented. The issue is neither discussed online nor in previous literature. Despite the popularity of LinTS and its applications, the susceptibility of its Python implementation to undesired non-deterministic behavior is not widely known. A general user who sets a specific seed for the pseudo-random generator, might expect deterministic results when no there are no inherent parallelism or asynchronous behavior. This user would remain oblivious to the fact that their work cannot be replicated across different environments. This is exactly what we would like to raise awareness for and address in this paper.

## 1.1 Our Contributions

First, we would like to bring attention to the non-deterministic behavior of LinTS which is not discussed in the original paper. This issue is concerned with LinTS implementation in Python. Therefore, it is not within the immediate scope of the original work. Our main contribution is to track down the root cause of non-determinism to Singular Value Decomposition (SVD) (Golub & Reinsch, 1970) of the covariance matrix when generating samples from the multivariate normal distribution.

As a remedy, we show that the issue can be avoided using Cholesky Decomposition (Gentle, 1998) instead of SVD, given that Cholesky's assumptions hold. We prove that using the Cholesky method is sound in the LinTS setting and then verify this numerically with extensive experiments across multiple environments[6].

---

[2]https://github.com/fidelity/mabwiser
[3]https://github.com/ntucllab/striatum
[4]https://github.com/Nth-iteration-labs/streamingbandit
[5]https://github.com/tensorflow/models/tree/archive/research/deep_contextual_bandits
[6]https://github.com/fidelity/mabwiser/tree/master/examples/lints_reproducibility

We also demonstrate that the DEEP BAYESIAN BANDITS paper (Riquelme et al., 2018), which proposes a new benchmark for bandits, suffer from the same reproducibility issue[7]. Based on our proposed approach, for reference, we contribute an open-source *deterministic* LinTS implementation in Python as part of our MABWISER library (Strong et al., 2021; 2019). Finally, we stress that using Cholesky decomposition is not the default setting in any NUMPY version when sampling from the multivariate normal distribution, and for best practice, we point out where special attention is required. This extends our contribution to other works beyond LinTS into broader recommender systems and IR community.

## 1.2 Organization of the Paper

We start with a brief background on bandits and LinTS in Section 2. Readers familiar with it can skip to Section 3, where we explain the non-deterministic behavior and its root cause. We then demonstrate the issue with a minimum example in Section 3.2. While non-determinism is a serious concern, thankfully, it is easy to address as we prove in Section 4. Finally, computational results in Section 5 quantify the impact of this issue on well-known recommendation benchmarks in terms of both raw scores and downstream evaluation metrics.

## 2 Background

In multi-armed bandits, an agent makes a sequence of decisions at time points $t = \{1, 2, \ldots, T\}$, whereby, at each time $t$, the agent chooses an arm $a_t$ from a set of $K$ arms. A single-arm is selected in the default case, while $k$ arms can be selected in *top-k*/slate recommendation setting (Swaminathan et al., 2017). After choosing one or more arms, a reward $r_t \sim R^{a_t}$ is observed for each selected arm from an unknown model of stochastic or deterministic outcomes. The rewards for other arms remain unknown. In contextual bandits, the agent also observes side information corresponding to the state of the environment at time $t$. This side information is referred to as *context* and is defined as $x_t \in \mathbb{R}^d$. The arm with the highest expected reward $\mathbb{E}[r_t|a_t]$ differs depending on this context. The uncertainty of the expected reward may differ based on context, which can impact decisions if the exploration strategy leverages that information.

In the Regret Minimization approach to optimizing bandit algorithms, the performance of a bandit algorithm is evaluated on the cumulative reward given by $r = \sum_{t=1}^{T} r_t$, or equivalently, on the cumulative regret incurred by the learning policy. If we define reward $r_t^*$ as the reward if the optimal action is chosen at time $t$, an optimal policy will generate a cumulative reward of $r^* = \sum_{t=1}^{T} r_t^*$, and the expected cumulative regret can be represented as: $L = \mathbb{E}[r^*] - \mathbb{E}[r]$. Overall, the objective of both context-free and contextual bandits is to maximize the cumulative reward, and equivalently, minimize the cumulative regret, which is the primary objective of the Regret Minimization task.

There exist a variety of learning policies that address the exploration-exploitation trade-off. The different strategies include frequentist ($\epsilon$-Greedy (Sutton & Barto, 2018), SoftMax (Sutton & Barto, 2018), UCB (Auer et al., 2002)) and Bayes-based (TS) approaches (Russo et al., 2018). In context-free bandits, the TS learning policy observes the number of successes and failures, generates a beta distribution for each arm, and samples from the beta to select the next arm. In contextual bandits, LinUCB (Li et al., 2010) and LinTS (Agrawal & Goyal, 2013) are the popular choices.

### 2.1 LinTS

The LinTS algorithm (Agrawal & Goyal, 2013) is the Bayes-based competitor of LinUCB. The main principle behind the LinTS approach is that the parameters used in ridge regression are sampled from multivariate normal distribution. This follows the Bayesian assumption that model parameters are not fixed and have prior distributions that can be revised upon observations. Algorithm 1 presents the details of Thompson Sampling with linear payoffs.

The LinTS algorithm starts by initializing a ridge regression for each arm. Notice that this uses the same $A_a^{-1} = (X_a^T X_a + \lambda I_d)^{-1}$ from LinUCB, with $\lambda$, again, as the regularization strength. In the initialization

---

[7]We plan to open issues in relevant repositories to share and rectify the problem.

---

**Algorithm 1:** Thompson Sampling with Linear Payoffs (LinTS)

---

Initialization for each arm $a$: $A_a = \lambda I_d$, $X_a^T r_a = 0^d$, $\beta_{\mu a} = 0^d$

**for** $t = 1, 2, ..., T$ **do**

    1. For each arm $a$, sample $\beta_{ta}$ from distribution $\mathcal{N}(\beta_{\mu a}, \alpha^2 A_a^{-1})$

    2. Select $arm_t = argmax_a(x_t \beta_{ta})$

    3. Observe reward $r_t$

    4. Update the selected arm:

        $A_a \leftarrow A_a + x_t^T x_t$

        $X_a^T r_a \leftarrow X_a^T r_a + x_t^T r_t$

        $\beta_{\mu a} \leftarrow A_a^{-1} X_a^T r_a$

---

step, there exist no $X_a$ observation, making the starting value of $A_a$ equal to $\lambda I_d$. The other components of the ridge regression, $X^T r$ and $\beta$ are initialized with zeroes.

At each time point $t$, a multivariate normal distribution is generated for each arm with the mean set as the current $\beta$ parameter of the arm, denoted by $\beta_{\mu a}$. For the covariance matrix, $M = \alpha^2 A_a^{-1}$ is used, where $\alpha$ is the exploration factor. A sampled $\beta_{ta}$ vector is drawn from this distribution. The non-deterministic behavior addressed in this paper occurs in this step (Step 1).

Next, the sampled values are used in the standard ridge regression calculation of $x_t \beta_{ta}$ to generate the expectation of the arm. The arm with the maximum expectation is selected (Step 2), and a reward $r_t$ is observed (Step 3). Based on the reward of the select arm, $A_a^{-1}$ and $\beta_{\mu a}$ are updated accordingly (Step 4). At time $t + 1$, and so on, the $\beta$ samples can be generated from revised multivariate normal distributions, and the algorithm proceeds.

## 3 Non-Deterministic Behavior of LinTS in Python

At inference time $t$ with a new context vector $x_t$, based on the trained parameters $\beta_\mu$ and $A$, the LinTS scoring procedure for each arm is as follows:

1. Find the inverse $A^{-1}$ of the trained model parameter $A$.

2. Scale $A^{-1}$ with the exploration parameter $\alpha^2$ to find the covariance matrix $M = \alpha^2 A^{-1}$.

3. Sample a $d$-dimensional vector $\beta_t$ from the multivariate normal distribution defined by the mean $\beta_\mu$ from the training and the covariance matrix $M$.

4. Find the dot product of the context vector $x_t$ with the sampled parameters $\beta_t$ to generate the likelihood.

In Python, a standard and straightforward implementation of this procedure uses the multivariate normal distribution functionality from the NUMPY package. Figure 1 outlines the corresponding implementation, which can be used to retrieve the score of a specific arm given the context vector $x_t$ (`x`) and the parameters $\beta_{\mu a}$ (`beta_mu`) and $A_a^{-1}$ (`A_inv`). The figure shows both the `RandomState` class usage (Option 1), which has been deprecated in July 2019 with version 1.18, and the new `Generator` class usage (Option 2). The multivariate normal distribution function provided in the `RandomState` generator uses SVD (Golub & Reinsch, 1970) to decompose the covariance matrix, which leads to non-determinism. Similarly, the new `Generator` class exclusively used SVD until the version 1.19 release in December 2019, which means that any implementation to that date has inherited the issue. To date, SVD remains the default option as the underlying decomposition technique in the multivariate normal distribution for both Option 1 and Option 2. In other words, any implementation similar to Figure 1 is problematic.

```
# Option 1) Random number generator - Legacy, Deprecated
rng = np.random.RandomState(seed=42)

# Option 2) Random number generator - New Generator class
rng = np.random.default_rng(42)

# Scoring function
def get_score_of_arm(x, beta_mu, A_inv, alpha=1):

    # Scaling with the exploration parameter
    M = np.square(alpha) * A_inv

    # Multivariate sampling
    beta_sampled = rng.multivariate_normal(beta_mu, M)

    # Score of the arm
    return np.dot(x, beta_sampled)
```

Figure 1: Python implementation of the LinTS scoring function for each arm in the multi-armed bandit.

### 3.1 The Root Cause Behind Non-Determinism

We narrow down the root cause of the issue to multivariate sampling. While it is not acknowledged in the NUMPY documentation, there exist few thread discussions in GitHub issues (#2435[8], #13597[9], #13386[10], and #13358[11]) that point to this underlying cause: the NUMPY multivariate sampling implementation guarantees the *distribution* of the samples but not the *values* of the samples.

Specifically, given the mean vector $m$ of dimension $d$ and the covariance vector $M$ of dimension $d \times d$, the `multivariate_normal` function generates new samples as follows:

1. Decompose the covariance matrix $M$ into a matrix $L$.

2. Randomly sample $d$ values from the standard normal distribution as $b$.

3. Add the mean vector $m$ to the dot product of the randomly sampled vector $b$ and the decomposed covariance matrix $L$ to give the final vector $s$.

How the matrix $L$ generated in the first step of the algorithm depends on the underlying decomposition method. As mentioned in Section 3, both the deprecated `RandomState` and the newer `Generator` classes use SVD.

The root cause of the non-determinism is, unfortunately, in the NUMPY's implementation of SVD. When two singular values are equal, they are returned in non-deterministic order. This subtle side effect is reported at least as early as 2012 in issue #2435. To make it worse, the issue is marked as "wont-fix" as the scope of this SVD implementation does not provide any guarantee on the values but only on the resulting distribution. As stated in issue #13358, the ordering of singular values is left to be determined by the underlying linear algebra library. Consequently, even when the seed value is fixed, sampling from a multivariate distribution with repeated singular values in the covariance matrix can lead to different results depending on the linear algebra library in the backend.

To confirm this behavior, we show that [12] with the exact same seed and NUMPY versions, the sampled values are in fact different depending on whether NUMPY is installed with Linear Algebra Package (LAPACK) (Anderson et al., 1999), Basic Linear Algebra Subprograms (BLAS) (Blackford et al., 2002), Open Source Basic Linear Algebra Subprograms (OpenBLAS) (Xianyi, 2011) or Intel Math Kernel Library (MKL) (Intel, 2003). Not only that, but due to numeric instability, there is no guarantee that results will remain reproducible even on the same machine with the same linear algebra backend.

---

[8]https://github.com/numpy/numpy/issues/2435
[9]https://github.com/numpy/numpy/issues/13597
[10]https://github.com/numpy/numpy/issues/13386
[11]https://github.com/numpy/numpy/issues/13358
[12]https://github.com/fidelity/mabwiser/tree/master/examples/lints_reproducibility

In the context of LinTS, the issue manifests itself with the mean vector $m \leftarrow \beta_\mu$, the covariance matrix $M \leftarrow \alpha^2 A^{-1}$, and the output $s \leftarrow \beta_t$. When the random sampling from the multivariate normal distribution is not guaranteed to be deterministic, then the inference made by LinTS is not reproducible. Notice that repeated singular values in the covariance matrix can naturally occur depending on the data.

Within the broader context of contextual bandits, it is important to be on the lookout for other algorithms that may be affected by the same issue. In general, UCB-based algorithms such as LinUCB (Li et al., 2010) and UCB-based Generalized Linear Bandits (Li et al., 2017; Kveton et al., 2020), which compute the optimal action without a stochastic component, are not susceptible to this non-determinism issue. On the other hand, TS-based contextual bandits such as PG-TS (Dumitrascu et al., 2018), and TS-based Generalized Linear Bandits such as GLM-TSL (Kveton et al., 2020) and SGD-TS (Ding et al., 2021) may be susceptible to non-determinism issues, as they include components that randomly sample from multivariate normal distribution.

In Section 5, our experimental analysis demonstrate that when the matrix of context vectors used in the training data is not linearly independent different scores are generated depending on the compute environment.

### 3.2 Minimal Example to Reproduce the Error

Let us finish this section with a minimal example to make the issue more concrete. Figure 2 presents toy training and test data sets with three arms.

```
# Three arms
arms = [1, 2, 3]

# Training data
decisions   = np.array([1, 1, 1, 2, 2, 3, 3, 3, 3, 3])
rewards     = np.array([0, 0, 1, 0, 0, 0, 0, 1, 1, 1])
contexts    = np.array([[0, 1, 2, 3, 5], [1, 1, 1, 1, 1],
                        [0, 0, 1, 0, 0], [0, 2, 2, 3, 5],
                        [1, 3, 1, 1, 1], [0, 0, 0, 0, 0],
                        [0, 1, 4, 3, 5], [0, 1, 2, 4, 5],
                        [1, 2, 1, 1, 3], [0, 2, 1, 0, 0]])
# Test data
test_context = np.array([0, 1, 2, 3, 5])

# Output
# >> [1, **2**, 3, 3, 3] (with MKL NumPy v1.18)
# >> [1, **1**, 3, 3, 3] (with OpenBLAS NumPy v1.18)
```

Figure 2: Example to reproduce non-deterministic LinTS.

We run this example using the LinTS implementation given in Figure 1 on the same operating system (Mac OS) with the same Python and NUMPY versions, and the same seed value for the random number generator. As can be seen in the results[13], the LinTS algorithm generates different results for the same context vector in the test data depending on the linear algebra packages MKL and OpenBLAS. Anecdotally, we encountered different results even with the same linear algebra package between two runs on the same machine. With larger number of arms, higher dimensional context vectors, and larger training and test data sets, we expect to see a greater frequency of such discrepancies. We demonstrate this effect further in our Computational Results section.

---

[13]https://github.com/fidelity/mabwiser/tree/master/examples/lints_reproducibility/
additional_experiments/lints_minimal

# 4 How to Guarantee Deterministic Thompson Sampling

The SVD implementation in NUMPY package returns singular values in *sorted* order. This suffers from instability when there are equal values. As a remedy, we need a technique that bypasses instability from pivoting or sorting operations and returns unique results. The Cholesky decomposition is one such method (Gentle, 1998). In the sections below, we show why Cholesky decomposition is an appropriate alternative to SVD within the context of the LinTS algorithm[14].

## 4.1 Cholesky Decomposition

The Cholesky decomposition operates over a Hermitian positive definite matrix and decomposes it into a lower triangular matrix and its conjugate transpose. It does not involve singular values or eigenvalues, follows a predictable set of steps with no pivoting or branching operation, and avoids the sorting problem. This yields deterministic behavior, up to numerical rounding errors, across different linear algebra packages. When reproducibility is of concern, Cholesky decomposition is one of the most recommended alternatives to SVD for multivariate normal sampling, see e.g. NUMPY GitHub issues #13358 and #13597.

The main requirement of Cholesky decomposition is its reliance on Hermitian positive definite matrices. In LinTS, the decomposition is applied to the covariance matrix $M = \alpha^2 A^{-1}$, for which we need to guarantee positive definiteness.

**LEMMA 4.1.** In LinTS, the covariance matrix $M = \alpha^2 A^{-1}$ is positive definite.

**PROOF.** We observe that before any training, the matrix $A$ is initialized as $A = \lambda I_d$, which is a real symmetric diagonal matrix, and therefore positive definite. During the training process, as shown in Algorithm 1, the matrix $A$ is updated using $x_t x_t^T$, where $x_t$ is a real-valued columnar vector, which leads to $x_t x_t^T$ being a real symmetric positive semidefinite matrix. This addition of $A$ and $x_t x_t^T$ has been shown to generate a positive definite matrix (Hoerl & Kennard, 1970), that is, the matrix $A$ remains positive definite during training. Since the inverse of a positive definite matrix is also positive definite (Horn & Johnson, 2012), the matrix $A^{-1}$ is also positive definite. The covariance matrix $M$ thus remains positive definite, making Cholesky decomposition appropriate for use during LinTS implementation.

Fortunately, in its recent versions, NUMPY introduced the option to select the decomposition method used for the multivariate normal sampling. The new `Generator` class provides this option in release 1.19 with Cholesky being one of the alternatives. We note that SVD remains to be the default option.

To ensure a deterministic implementation of LinTS in Python, we recommend the following:

1. The multivariate normal function in the legacy `RandomState` number generator (Option 1) should not be used in LinTS implementations since it uses SVD exclusively. If `RandomState` must be used, e.g., for backwards compatibility, we recommend a custom implementation of the sampling function.

2. The new `Generator` (Option 2) is suitable for LinTS. The important distinction is that instead the default SVD option, Cholesky should be selected when sampling from the multivariate normal distribution.

```
# Make sure to select Cholesky decomposition
beta_sampled = rng.multivariate_normal(beta_mu, M, method='cholesky')
```

Figure 3: Cholesky decomposition for deterministic multivariate normal sampling.

Our running implementation illustrated in Figure 1 can be modified to adhere to these guidelines as in Figure 3. With this minor update, the minimal example in Figure 3.2 now returns identical results. The adverse impact of not following these suggestions is demonstrated next.

---

[14]We would also like to acknowledge our action editor Lihong Li's suggestion of adding a small noise to the initialization of the matrix $A$. Our initial empirical findings confirm that an appropriately sized noise deterministically applied to the diagonal of the matrix $A$ at the time of initialization can be another potential solution.

| Operating System (OS) | OS Version | NumPy LA Backend |
|:---:|:---:|:---:|
| Windows | 10.10 | OpenBLAS |
| Red Hat Linux | 3.10 | OpenBLAS |
| MacOS Darwin | 10.15 (19.6) | OpenBLAS |
| MacOS Darwin | 10.15 (19.5) | MKL |
| Amazon AWS Linux | 4.14 | LAPACK |

Table 1: Environments considered in our experiments with varying operating systems and linear algebra backend.

## 5 Computational Results

In the previous sections, we explored non-determinism in LinTS, revealed the root cause behind the problem, and proposed a simple solution with easy-to-follow best practices. What remains to be seen is a computational study to quantify the impact of the non-deterministic behavior and understand the extent it impacts downstream tasks and metrics. In particular, the goal of our experiments is to answer two main questions:

[**Q1**] What is the similarity between the results produced by a non-deterministic implementation of the LinTS algorithm under different environment setups? Does a deterministic implementation resolve this dissimilarity?

[**Q2**] Does the issue manifest itself in standard benchmarks and existing Python libraries, and what is its impact on relevant evaluation metrics?

To answer these questions, we consider Recommendation System as our application domain, where LinTS is commonly used, and conduct experiments on the well-known MovieLens data set (Harper & Konstan, 2016). As a negative result, we observe that when using SVD, the recommendations exhibit considerable discrepancies in both generated item scores and recommendation metrics. Practitioners might attribute such performance differences incorrectly to algorithmic improvements while it is solely due to non-determinism.

As a positive result, we find that using Cholesky yields identical results across all environments, independent of the OS and the linear algebra backend. This confirms our proof experimentally and verifies the effectiveness of our proposed solution. Finally, we focus on Google Brain's Deep Bayesian Bandits library (Riquelme et al., 2018) to test a 3rd party implementation, confirming it also suffers in terms of reproducibility.

### 5.1 Environment Setup

Table 1 provides the list of environments considered in our experiments. We target mainstream operating systems Windows, Linux, and Mac OS. We also include Amazon AWS Linux as an example of a cloud service provider. In parallel, we consider three different linear algebra backends: Open Source Basic Linear Algebra Subprograms (OpenBLAS) (Xianyi, 2011), Linear Algebra Package (LAPACK) (Anderson et al., 1999) and Intel Math Kernel Library (MKL) (Intel, 2003). Note that even with the same underlying linear algebra package, there can be differences between operating systems.

### 5.2 Numerical Results on MovieLens Data

We use the well-known 100K MovieLens data set (Harper & Konstan, 2016) with 100K ratings from 943 users for 1682 movies. For TS, we binarize the data set for each user-movie pair with a positive reward if the user rated the movie and zero otherwise. This gives us 1.58M observations with 100K positive responses. We divide it into 70%-30% train and test splits. The user context includes age, gender (boolean), occupation (one-hot), and the ZIP code (ordinal). We augment this context with a column representing an unknown occupation (*none* or *other*). This minor feature engineering creates linear dependence in the data and duplicate singular values in $A^{-1}$ matrix.

| Env | Comparison | | | | |
|---|---|---|---|---|---|
| Red Hat Linux | Score | 0% | | | |
| | Prediction | 32.64% | | | |
| MacOS MKL | Score | 0.00021% | 4.2e-4% | | |
| | Prediction | 43.87% | 67.36% | | |
| MacOS OpenBLAS | Score | 0% | **100%** | 0.00042% | |
| | Prediction | 32.64% | **100%** | 67.36% | |
| AWS Linux | Score | 0% | **100%** | 0.00042% | **100%** |
| | Prediction | 32.64% | **100%** | 67.36% | **100%** |
| | | Windows | Red Hat Linux | Mac MKL | MacOS OpenBLAS |

Table 2: Numerical Results on MovieLens: Comparison of the percentage of matching raw user-item scores and matching predictions on whether to recommend the item across environment pairs when using SVD-based LinTS implementation. A reproducible result should match 476,006 (100%) times as shown in **bold**. Any other result is a discrepancy due to non-deterministic behavior.

| Environment | CR | NDCG | | | Precision | | | Recall | | |
|---|---|---|---|---|---|---|---|---|---|---|
| | | @5 | @10 | @25 | @5 | @10 | @25 | @5 | @10 | @25 |
| Windows | 37 | 0.0166 | 0.0244 | 0.0399 | 0.1251 | 0.1208 | 0.1102 | 0.0064 | 0.0118 | 0.0265 |
| Red Hat | 54 | 0.0201 | 0.0286 | 0.0457 | 0.1399 | 0.1300 | 0.1175 | 0.0070 | 0.0132 | 0.0297 |
| MacOS MKL | 40 | 0.0181 | 0.0270 | 0.0418 | 0.1336 | 0.1265 | 0.1124 | 0.0071 | 0.0132 | 0.0271 |
| Mac OpenBLAS | 54 | 0.0201 | 0.0286 | 0.0457 | 0.1399 | 0.1300 | 0.1175 | 0.0070 | 0.0132 | 0.0297 |
| AWS Linux | 54 | 0.0201 | 0.0286 | 0.0457 | 0.1399 | 0.1300 | 0.1175 | 0.0070 | 0.0132 | 0.0297 |
| Cholesky (for all) | 31 | 0.0131 | 0.0190 | 0.0312 | 0.0926 | 0.0905 | 0.0858 | 0.0048 | 0.0090 | 0.0203 |

Table 3: Numerical Results on MovieLens Data: Comparison of evaluation metrics at different top-k settings across different environments when using SVD-based LinTS implementation. The last row shows the Cholesky implementation that is identical for all environments.

Table 2 presents our results that compare all pairs of environments on the MovieLens data using the SVD-based LinTS implementation. The table contains the matching percentages for the raw user-item scores generated by LinTS, and whether both environments made the same prediction on recommending the item to the user. Ideally, we expect 476,006 matches, as shown in bold. We find that this is the case for three pairs of environments, which match 100%, while the others have an low percentage of predictions match and very few matches in scores. Let us note, compared to our variant of the MovieLens dataset studied here, real-world datasets might capture richer structure that could lead to even worse discrepancies.

Table 3 presents the results for recommendation metrics for different top-k recommendations from 5 to 25, and the cumulative reward (CR) when only a single arm is selected. The cumulative reward is calculated as the sum of rewards across the recommendations of the model, and recommendation metrics follow their standard definitions (Sun et al., 2020). As can be seen, the failure of matching scores propagates downstream and causes significant differences in the evaluation metrics. The NDCG@5 difference between Windows and MacOS OpenBLAS environments is enough to change the outcome of an experiment, leading to incorrect conclusions. When using Cholesky decomposition instead, as we expect from the theory, all environments return the same scores 100% of the time, leading to identical metrics.

| OS | Backend | RS | SVD | Cholesky |
|---|---|---|---|---|
| MacOS Big Sur | MKL | 4135 | 3575 | **4140** |
| MacOS Big Sur | OpenBLAS | 3720 | 3830 | **4140** |
| Ubuntu 18.04 | MKL | 3475 | 3540 | **4140** |
| Ubuntu 18.04 | OpenBLAS | 4075 | 3875 | **4140** |

Table 4: Numerical Results with the Deep Bayesian Bandit Library: Comparison of the cumulative reward across different environments when using the SVD-based and Cholesky-based LinTS implementation. The identical reward are shown in **bold**. Any other result is a discrepancy due to non-deterministic behavior.

### 5.2.1 Additional Experiments:

We conducted further experiments[15] on other recommendation benchmarks such as the GoodReads data set (Wan & McAuley, 2018) with review-based user features generated by TextWiser (Kilitcioglu & Kadioglu, 2021), on MovieLens with parameter tuning, and on a synthetic dataset. All of these additional experiments lead to the same conclusion and are omitted here.

### 5.3 Numerical Results with the Deep Bayesian Bandits Library

As our final experiment, we turn our attention to an existing library and test an implementation aside from our own, focusing on the Deep Bayesian Bandits library from Google Brain (Riquelme et al., 2018).

This library contains the LinTS algorithm to account for its baseline approach. The LinTS implementation[16] of the paper (Riquelme et al., 2018) makes a sampling call to the `multivariate_normal` function as we warned against in Section 3. Crucially, the LinTS algorithm shows the best cumulative regret for the UCI mushroom data set(Dua & Graff, 2017) in this paper (Riquelme et al., 2018), which calls for verification.

Many thanks to our authors who provided example code to reproduce their table, we were able to run their code to test whether this LinTS algorithm is prone to reproducibility issues[17]. We use the UCI mushroom data set(Dua & Graff, 2017) in a bandit task (Blundell et al., 2015), taking a sample of 2000 to fully mimic the example. We then compare the reproducibility across three different approaches:

- `RandomState` (RS): The legacy option from Figure 1 which uses SVD. The Deep Bayesian Bandits library implements this option. We plan to reach out to the contributors of this library to share our findings.

- `SVD`: The newer `Generator` class for random number generator that uses the default SVD decomposition, as shown in Section 3.

- `Cholesky`: The newer `Generator` class that explicitly specifies the Cholesky decomposition (Figure 3). This is the suggested method of our paper.

To measure the difference, we use the *cumulative reward* generated by the LinTS policy, which is used to select the best performing policy when comparing bandit algorithms. It is therefore critical that the rewards are reproducible.

Table 4 presents the results which show that the cumulative reward differs considerably across different environments for SVD-based implementations. Once again, we show that using the Cholesky decomposition alleviates the problem completely, producing the same reward across all environments. Note that in contrast to the recommendation metrics in the previous example, Cholesky leads to better performance in this task.

---

[15]https://github.com/fidelity/mabwiser/tree/master/examples/lints_reproducibility/additional_experiments

[16]https://github.com/tensorflow/models/blob/archive/research/deep_contextual_bandits/bandits/algorithms/linear_full_posterior_sampling.py#L98

[17]https://github.com/fidelity/mabwiser/tree/master/examples/lints_reproducibility/table_4

This also calls the existing results into question. Our findings indicate that the previous results will not be reproducible even if we were to use their exact implementation. We reiterate our recommendation for using the Cholesky decomposition to ensure deterministic LinTS behavior, which also applies to Google's Deep Bayesian Bandits (Riquelme et al., 2018) library as well as any academic or industrial work that depends on it.

## 6 Conclusion

Almost ten years after the impactful LinTS paper (Agrawal & Goyal, 2013), we presented a detailed treatment of the subtle non-deterministic behavior that jeopardizes its implementations in Python. LinTS offers great versatility in quantifying uncertainty and provides a principled mechanism for exploration. However, attention is required to guarantee deterministic behavior as we advocated in this paper.

We identified the root cause of the issue, discussed potential sources of instability that can lead to unwanted behavior, provided a minimal example and multiple experiments to reproduce the error, and studied it in details. Our experiments demonstrated that the issue manifests itself in well-known recommendation benchmarks, impacts existing libraries and any work that utilizes NumPy's multivariate sampling implementation, and can be detrimental to computational results and conclusions drawn. Finally, and most importantly, we described a simple fix based on Cholesky decomposition to rectify the issue within the context of the LinTS algorithm. We proved the soundness of the technique and shared general advice with practitioners and researchers.

Our open-source contextual multi-armed bandit library, MABWiser (Strong et al., 2021; 2019) is released with this deterministic LinTS implementation as a reference. Overall, our findings take a step toward reproducible AI research and hopefully motivate others not to overlook such important issues as forgotten implementation details that might influence scientific conclusions.

### Acknowledgments

We would like to thank Anshuman Pradhan and Pramod R for bringing this issue to our attention, and to Emily Strong for providing the initial implementation in MABWiser (Strong et al., 2021; 2019). We would also like to thank our anonymous reviewers and our action editor Lihong Li for their reviews and feedback which has helped improve this work.

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
