# OpenReview forum: "Non-Deterministic Behavior of Thompson Sampling with Linear Payoffs and How to Avoid It"
_TMLR — Accepted by TMLR_

### Review · Reviewer_Sa2S · 2022-05-09

**Summary Of Contributions:**

This work focuses on the non-deterministic behavior of the Lin-TS algorithm, which is claimed to be caused by the SVD calculation in the application of Lin-TS (the step involving multivariate normal distribution function in particular). It studies this topic by exploring how the general tool, Python, implements each step of the algorithm and suggests that this issue may be addressed by replacing SVD with Cholesky decomposition.

**Broader Impact Concerns:**

No concerns on the ethical implications of the work in my view.

**Requested Changes:**

In general, I believe this work is interesting and worth attention. However, I suggest some changes to convince me the claims are reasonable as detailed in the 'Strengths And Weaknesses' part:
1. In Section 2, a more detailed introduction of the background is appreciated, which will also help the readers to understand how the performance of LinTS is evaluated in the experiment section (LinTS was originally proposed for regret minimization as I know).
1. In Section 5, more explanation on the results may help understanding. I do believe the conclusions are reasonably drawn from the experiment results, but currently it is a bit difficult for me to understand the tables now.

**Strengths And Weaknesses:**

Strengths:
1. This work explores how Python will implement the Lin-TS algorithm in great detail and also provide extensive experiment results. This topic is interesting and worth to explore as it will help scholars to develop a better understanding about the implication of TS algorithms.
1. It is well-organized and generally easy to read.

Weaknesses:
1. Section 2
   1. The first paragraph mentions `top-k recommendation', which seems to relate to the best arm identification/pure exploration (BAI) in multi-armed bandits. However, the second paragraph simply suggests that 'The performance of a bandit algorithm is evaluated on the cumulative reward'. (Regret Minimization/RM)
   1. In general, BAI and RM are two common targets in the field of multi-armed bandits. I think this section should describe and distinct these two targets and provide more references to relating works.
   1. I understand this work is mainly about the experiments of LinTS (a RM algorithm), but a clear description on the context is still appreciated. A clear introduction of multi-armed bandits can be found in many existing works such as the book ['Bandit Algorithms'](https://banditalgs.com/).
   1. Besides, the notation $E[r|a]$ is used in the first paragraph but the definition of neither 'r' nor 'a' is clearly explained.
1. Section 5
   1. I can understand the claim of the authors from the paragraphs, but the results look confusing to me since I don't quite understand how the comparison is done. Is the cumulative regret applied to evaluate the performance of the algorithm?
   1. How many times have the same setup been simulated?
   1. What is the meaning of 'Score', 'Probability', and 'Prediction' in Table 2?
   1. May you provide more explanations on the evaluation metrics and environment setups in Table 3? What do 5, 10, and 25 stand for?

---

### Review · Reviewer_5XTG · 2022-05-10

**Summary Of Contributions:**

This work investigates the usage of the well-known Linear Thompson Sampling algorithm in practice. The focus is in evaluating its reproducibility due to its randomness. Authors showed that the original algorithm is actually characterized by a non-deterministic behavior preventing its usage for industrial applications.

**Broader Impact Concerns:**

I did not find any ethical concern with this work.

**Requested Changes:**

I do not have specific changes to request.

**Strengths And Weaknesses:**

The writing is clear and well organized. The technical problem is explained in a clear way and is supported properly. I don't have clear suggestions for improving this work.

That said, I consider that the brought contribute might be of benefit for researchers in the bandit area. Hence, as both the technical part seems correct and well supported, and there is a contribute given by this work I am satisfied with it.

---

### Review · Reviewer_bVwk · 2022-05-23

**Summary Of Contributions:**

This paper points out a reproducibility issue/bug of many popular and widely used implementations of the LinTS (Thompson Sampling with Linear Payoffs) that are based on MumPy's  implementation of SVD (singular value decomposition). LinTS is a fundamental and widely used machine learning algorithm for solving the contextual bandit problems, and has found applications in various areas such as recommender systems and ranking. This but is that: in some implementations of LinTS, multiple experiments with the same random seed, the same Python codes, and the same datasets could return different results, which may cause the results of previous experiments unable to reproduce. The authors also did experiments on real-world movie datasets to show this bug. The root cause of the bug comes from NumPy's implementation of SVD that many LinTS implementations base on. In NumPy's implementation of SVD, when the matrix has two or more equal singular values, then there order in the return results of SVD are not deterministic and may be affected by the devices/systems. Therefore, the LinTS implementations based on it will also be non-deterministic. Then, the authors introduced Cholesky Decomposition for replacing the SVD, which does not have such bug. The authors also provide a LinTS implementation based on it, The authors show that this new implementation always returns the same result in different systems/devices, and thus, fixed this bug.

**Broader Impact Concerns:**

This paper does not have ethical concerns.

**Requested Changes:**

Resolve the weakness 1 and 2. Resolving weakness 3 is optional but can well strengthen this paper.

For the current version, I will vote a major revision and want to see weakness 1 and 2 addressed.

**Strengths And Weaknesses:**

Strengths
It is exciting and inspiring to find a bug that is wide and common. This paper can help prevent reproducibility issues and reduce the engineer efforts for tuning models/parameters, and may have helped some. I personally have seen some similar reproducibility problems in some industry machine learning systems and people were struggling finding a solution. I appreciate the authors to point out this bug and provides a solution.

Weakness
1. It is unclear how severe this bug is. The authors do not provide enough evidence to show the severity. For the experiments, it is not clear how this non-deterministic issue caused more severe problems than other possible randomness. For instance, in some distributed machine learning algorithms (eg EASGD, an asynchronized optimization algorithm), the completion order of CPUs/GPUs/nodes is random and may also lead to non-deterministic behaviors of the trained models. The authors should be more clear on this point.
2. This paper needs to justify whether and why Cholesky Decomposition is a better solution compared to making a deterministic implementation of SVD. The content of this paper does not convince me that we should use Cholesky to replace a widely-used and supported algorithm SVD.
3. Maybe an adversarial example that escalates this bug may help strengthen this paper.

---

### Decision · Action_Editors · 2022-07-01

**Recommendation:** Accept with minor revision

**Comment:**

The paper studies a reproducibility issue in popular implementation of LinTS based on NumPy's SVD implementation. Given wide application of LinTS and popularity of NumPy, reviewers find the problem and solution to be relevant and useful to the community. They also agree that the solution is technically sound and well supported, and that the writing is of good quality. The responses also answered some of the questions in initial reviews. All reviewers are positive about this work, so I'd like to recommend Accept with Minor Revision.

In addition to what reviewers have pointed out, I'd like to add two suggestions, and look forward to responses in the revision:

First, nondeterminism happens when two singular values are the same. Would it be an easier fix to add a small, deterministic perturbation to diagonal elements when initializing matrix A (first line of Algorithm 1)? If the perturbation is different in element, and the magnitude is small, we will get almost identical performance as unperturbed LinTS, and the chances of seeing non-deterministic results are essentially 0.

Second, while the paper focuses on LinTS, can the authors have a discussion of whether nondeterminism happens in other similar algorithms, such as LinUCB, and TS with generalized linear models, etc.? While the answers may be straightforward in some cases, it'll still be helpful to make them explicit.

---

> ### Author Response · Authors · 2022-07-05
> **Response to decision and suggestions by the action editor**
>
> We would like to thank our action editor for their suggestions and their recommendation.
>
> Regarding the first suggestion, our initial findings confirm our action editor's great intuition. We wish to include this suggestion in our paper and our supplementary material, attributing to our action editor. Please let us know if you do not want this suggestion to be included in the paper.
>
> Regarding the second suggestion, we will add a discussion in our paper to further clarify the impact of this nondeterminism issue on similar algorithms.
>
> We will follow-up with the camera-ready revision that includes these additions.

---

> > ### Comment · Action_Editors · 2022-07-05
> > **Thank you for following up**
> >
> > Thank you for following up. We look forward to the new version.